# Deep phenotyping of 34,128 adult patients hospitalised with COVID-19 in an international network study

Edward Burn ◉ et al.[#]

Comorbid conditions appear to be common among individuals hospitalised with coronavirus disease 2019 (COVID-19) but estimates of prevalence vary and little is known about the prior medication use of patients. Here, we describe the characteristics of adults hospitalised with COVID-19 and compare them with influenza patients. We include 34,128 (US: 8362, South Korea: 7341, Spain: 18,425) COVID-19 patients, summarising between 4811 and 11,643 unique aggregate characteristics. COVID-19 patients have been majority male in the US and Spain, but predominantly female in South Korea. Age profiles vary across data sources. Compared to 84,585 individuals hospitalised with influenza in 2014-19, COVID-19 patients have more typically been male, younger, and with fewer comorbidities and lower medication use. While protecting groups vulnerable to influenza is likely a useful starting point in the response to COVID-19, strategies will likely need to be broadened to reflect the particular characteristics of individuals being hospitalised with COVID-19.

---

[#]A list of authors and their affiliations appears at the end of the paper.

The ongoing coronavirus disease 2019 (COVID-19) pandemic is placing a huge strain on health systems worldwide. While a number of studies have provided information on the clinical characteristics of individuals being hospitalised with COVID-19[1–3], substantial uncertainty around the prevalence of comorbidities and prior medication use among this population remains. Moreover, it is not known whether those hospitalised with COVID-19 are systematically different from individuals hospitalised during previous influenza seasons.

COVID-19 shares similarities with influenza to the extent that both cause respiratory disease which can vary markedly in its severity and present with a similar constellation of symptoms, including fever, cough, myalgia, malaise, fatigue and dyspnoea. Early reports do, however, indicate that the proportion of severe infections and mortality rate is higher for COVID-19[4]. Older age and a range of underlying health conditions, such as immune deficiency, cardiovascular disease, chronic lung disease, neuromuscular disease, neurological disease, chronic renal disease and metabolic diseases, have been associated with an increased risk of severe influenza and associated mortality[5]. While age appears to be a clear risk factor for severe COVID-19[4], other associations are not yet well understood. Comparisons with COVID-19 are further complicated by the heterogeneity in influenza itself, with different strains resulting in different clinical presentations and associated risks. Those hospitalised with the A(H1N1) pdm09 subtype of the influenza A virus during the associated influenza pandemic in 2009 were, for example, generally younger and with fewer comorbidities than those from preceding influenza seasons[6].

Here we first aimed to describe the characteristics of patients hospitalised with COVID-19. In particular, we set out to summarise individuals' demographics, medical conditions, and medication use. Our second aim was to compare the characteristics of individuals hospitalised with COVID-19 to those of patients hospitalised with influenza in previous seasons.

## Results

**Patients hospitalised with COVID-19**. A total of 34,128 individuals hospitalised with COVID-19 (CUIMC: 1759; HIRA: 7341; HM: 2078; PHD: 5257; SIDIAP: 16,347; UC HDC: 769; VA OMOP: 577) were included. In all, 68,829 summary characteristics, from 4811 (HM) to 11,643 (CUIMC) unique aggregate characteristics, were extracted and summarised. All are made available in an accompanying interactive website (http://evidence.ohdsi.org/Covid19CharacterizationHospitalization/).

Cohorts from CUIMC, HM, PHD, SIDIAP, UC HDC and VA OMOP were predominantly male (52%, 60%, 52%, 54%, 54% and 94%, respectively, but mostly female for HIRA (60%). The age distributions of those hospitalised for COVID-19 are summarised in Fig. 1 (alongside those hospitalised with influenza, see below). Different patterns are seen in the various contributing databases with, in particular, patients in South Korea (HIRA) seen to be younger than elsewhere.

The distribution of comorbidities in COVID-19 patients varied across sites and countries, see Table 1. The mean Charlson comorbidity index of those hospitalised with COVID-19 in the US ranged from 3.1 for PHD to 5.4 for VA OMOP, from 0.8 for HM to 1.4 for SIDIAP in Spain, and was 2.7 in HIRA, covering South Korea. In the US, the proportion of those hospitalised with COVID-19 who had asthma ranged from 7 to 15%, from 24 to 43% for diabetes, from 28 to 49% for heart disease and from 8 to 18% for cancer. In Spain, between 4 and 7% for asthma, from 13 to 22% for diabetes, from 17 to 27% for heart disease and from 5 to 16% for cancer. In South Korea, 12% of those hospitalised had a history of asthma, 17% had diabetes, 13% heart disease and 4%

cancer. The prevalence of hypertension ranged from 37 to 70% in the US, from 30 to 46% in Spain and was 24% in South Korea. The prevalence of the full range of conditions summarised are shown in Fig. 2, with all values reported at http://evidence.ohdsi.org/Covid19CharacterizationHospitalization/.

For medications, the proportion of those hospitalised with COVID-19 in the US who had been taking agents acting on the renin–angiotensin system over the 30 days up to their hospitalisation ranged from 18% to 39%, while the proportions taking immunosuppressants ranged from 4 to 6%, and from 21 to 51% for lipid-modifying agents over the same time period. In HIRA, 14% had been taking agents acting on the renin–angiotensin system, 1% immunosuppressants, and 16% lipid-modifying agents. In SIDIAP, 27% had been taking agents acting on the renin–angiotensin system, 2% immunosuppressants, and 24% lipid-modifying agents (Table 2). Looking at one medication of particular interest (of which many can be explored in detail at http://evidence.ohdsi.org/Covid19CharacterizationHospitalization/), use of hydroxychloroquine on the day of admission was 7% in HIRA and 41% in HM. The prevalence of the many medications summarised are shown in Fig. 2, with all values reported http://evidence.ohdsi.org/Covid19CharacterizationHospitalization/.

Removing the requirement of having a year prior history in datasets other than HM and Premier did not materially change the results (see http://evidence.ohdsi.org/Covid19CharacterizationHospitalization/ for full details).

**A comparison of patients hospitalised with COVID-19 and patients hospitalised with influenza**. A total of 84,585 patients hospitalised with influenza between 2014 and 2019 (2125 CUIMC, 49,977 PHD, 2947 SIDIAP, UC HDC 26,547 VA OMOP). In addition, 2443 patients hospitalised with influenza between 2009 to 2010 were included (170 CUIMC, 1689 SIDIAP, 584 VA OMOP) were also identified. Patient characteristics of those hospitalised with COVID-19 are compared to those of individuals hospitalised with influenza between 2014 and 2019 in Figs. 1, 3 and 4, and with those hospitalised with influenza between 2009 and 2010 in Supplementary Fig. 1 and Supplementary Fig. 2.

A greater proportion of those hospitalised with COVID-19 were male compared to those hospitalised with influenza between 2014 and 2019 for CUIMC, PHD, SIDIAP and UC HDC. Of those hospitalised between 2014 and 2019 with influenza 43, 44, 53 and 45% were male for each of these respective data sources. The ages of those hospitalised with COVID-19 generally appeared slightly younger compared to those hospitalised with influenza between 2014 and 2019, see Fig. 1.

Those hospitalised with COVID-19 had a comparable or lower prevalence of comorbidities compared to those hospitalised with influenza 2014–2019 (see Fig. 3 top and Fig. 4). Chronic obstructive pulmonary disease (COPD), cardiovascular disease and dementia were all more common amongst those hospitalised with influenza compared to those hospitalised with COVID-19. Medication use was less common amongst COVID-19 patients (Fig. 3 bottom and Fig. 4) with, for example, systemic corticosteroids and alpha-blockers amongst those more frequently used among those hospitalised with influenza.

Those hospitalised with influenza between 2009 and 2010 were typically younger compared to both COVID-19 and to influenza 2014–2019 admissions (see Supplementary Fig. 1 and Supplementary Table 2). COVID-19 patients were more likely to be male, with 40, 49 and 91% of those hospitalised between 2009 and 2010 for influenza being male for CUIMC, SIDIAP and VA OMOP. Comparisons of conditions and medications, however,

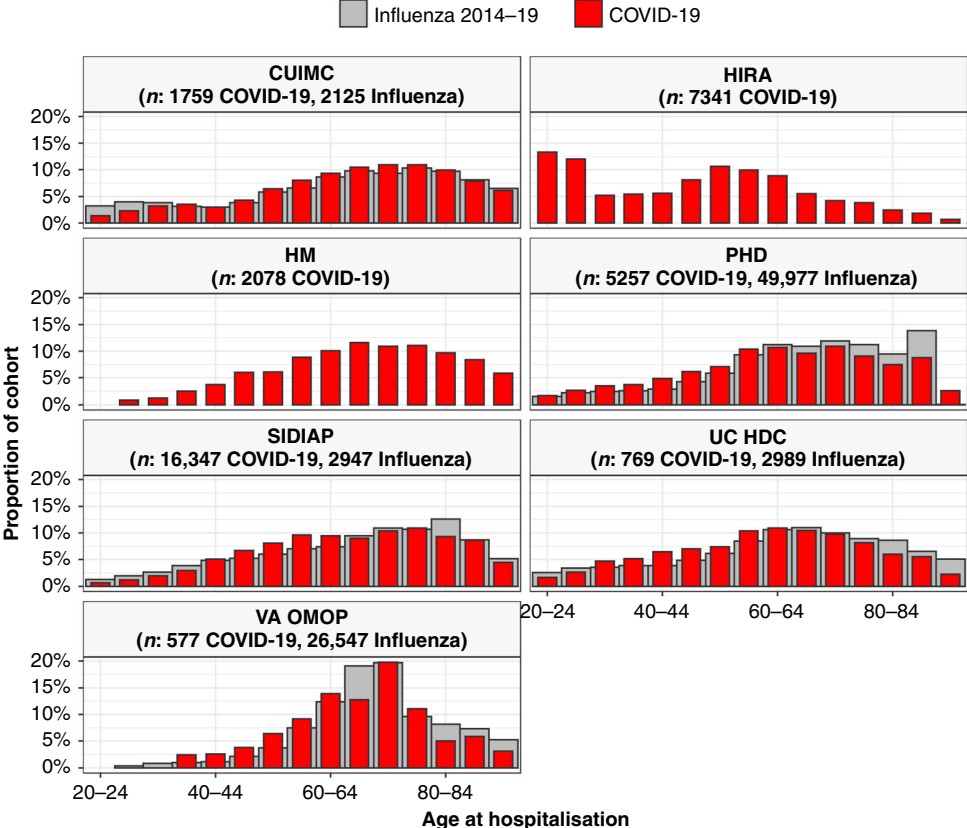

**Fig. 1 Age of patients hospitalised with COVID-19 and of patients hospitalised with influenza.** Individuals hospitalised with COVID-19 between December 2019 and April 2020 compared with those hospitalised with influenza between September 2014 to April 2019 (where available). Proportion of cohorts by 5-year age groups, with groups with counts of <10 omitted. CUIMC: Columbia University Irving Medical Center; HIRA: Health Insurance Review & Assessment; HM: HM Hospitales; PHD: Premier Healthcare Database; SIDIAP: The Information System for Research in Primary Care; UC HDC: University of Colorado Health Data Compass; VA OMOP: Department of Veterans Affairs. Influenza data for SIDIAP was only available from 2014 to 2017.

varied depending on the data source (see Supplementary Fig. 4 and Supplementary Table 2).

## Discussion

**Summary of key findings**. The characteristics of 34,128 patients hospitalised with COVID-19 in US, South Korea, and Spain have been extracted from EHRs and health claims databases and summarised. Between 5000 and 12,000 unique aggregate characteristics have been produced across databases, with all made publicly available in an accompanying website.

Patients hospitalised with COVID-19 in the US and Spain were predominantly male with age distributions varying across data sources, while those in South Korea were mostly female and appreciably younger than patients in the US and Spain. Many comorbidities were common among individuals hospitalised with COVID-19 with, as an example, 37–70% of those hospitalised with COVID-19 in the US, 30–46% of those in Spain and 24% of those in South Korea having hypertension. Similarly, prior medication use was common with, for example, 18–39% in the US, 27% in Spain and 14% in South Korea, taking drugs acting on the renin–angiotensin system (ACE inhibitors and ARBs) in the 30 days up to their hospitalisation.

Comparisons with previous cohorts of patients admitted to hospital for seasonal influenza in recent years suggest that COVID-19-related admissions are seen more often in younger patients and with a higher proportion of men. In the US and Spain, those hospitalised with COVID-19 were generally either of comparable health or healthier than patients hospitalised with influenza. Consistent differences were noted in the prevalence of

respiratory disease, cardiovascular disease and dementia, each more common amongst those hospitalised with influenza in all of the contributing databases. Similarly, the use of corticosteroids and alpha-blockers was consistently higher amongst influenza patients. Those hospitalised with influenza in 2009–2010, during the pandemic associated with H1N1, were seen to be younger than both those hospitalised with influenza in more recent years and patients hospitalised with COVID-19.

**Findings in context**. A number of studies have previously provided information on individuals hospitalised with COVID-19. While cohorts have generally been majority male, the prevalence of comorbidities has varied. In a study of 1099 individuals who tested positive for COVID-19 in China, of whom 94% were hospitalised, 58% were male, with 7% having diabetes, 15% hypertension and 1% cancer[7]. In another study of 191 patients with COVID-19 in two hospitals in Wuhan, China, 62% were male, 19% had diabetes, 30% had hypertension and 1% had cancer[8]. In a study which identified 1999 individuals who tested positive for COVID-19 and were hospitalised in New York, 63% were male, 25% had diabetes, 10% COPD and 45% a cardiovascular condition[9]. In another US study of 1482 patients admitted to hospital with COVID-19 in March 2020, 55% were male, with 28% having diabetes, 11% having COPD and 28% having cardiovascular disease[10]. Our findings add to this emerging body of evidence. The results from our study also provide an illustration of the variation in patient characteristics across contexts, with heterogeneity seen both across the cohorts from the US and between the US, Spain and South Korea.

**Table 1 Conditions of individuals hospitalised with COVID-19.**

|  | CUIMC (n: 1759) | HIRA (n: 7341) | HM (n: 2078) | PHD (n: 5257) | SIDIAP (n: 16,347) | UC HDC (n: 769) | VA OMOP (n: 577) |
|---|---|---|---|---|---|---|---|
| Charlson index | 4.0 | 2.0 | 0.8 | 3.1 | 1.5 | 3.5 | 5.4 |
| Anaemia | 13.1% | 11.9% | 4.2% | 30.2% | 14.6% | 29.9% | 30.0% |
| Anxiety disorder | 5.5% | 11.5% | 2.3% | 18.0% | 23.7% | 18.2% | 30.3% |
| Asthma | 7.2% | 12.0% | 4.2% | 14.6% | 6.9% | 13.8% | 9.9% |
| Atrial fibrillation | 8.4% | 1.0% | 6.8% | 18.1% | 8.2% | 12.0% | 15.8% |
| Chronic liver disease | 2.6% | 5.0% | 0.6% | 3.0% | 1.9% | 2.3% | 6.8% |
| COPD | 6.1% | 1.9% | 5.9% | 24.5% | 7.6% | 12.6% | 28.6% |
| Dementia | 8.5% | 5.3% | 2.3% | 9.2% | 5.6% | 7.4% | 7.8% |
| Diabetes mellitus | 24.2% | 16.6% | 13.2% | 35.8% | 21.6% | 35.2% | 43.3% |
| GERD | 9.6% | 32.3% | 1.6% | 24.8% | 9.5% | 23.4% | 26.9% |
| Heart disease | 28.3% | 13.1% | 16.5% | 48.6% | 27.1% | 39.0% | 48.2% |
| Heart failure | 10.7% | 5.1% | 2.7% | 24.8% | 5.9% | 12.9% | 22.2% |
| Hyperlipidemia | 24.7% | 31.1% | 21.3% | 46.0% | 23.7% | 38.0% | 55.8% |
| Hypertensive disorder | 36.6% | 23.8% | 29.6% | 47.1% | 45.6% | 57.3% | 69.7% |
| Insomnia | 3.1% | 4.2% | 0.7% | 4.4% | 14.6% | 7.4% | 10.9% |
| Ischaemic heart disease | 8.0% | 6.9% | 4.1% | 16.0% | 7.3% | 13.5% | 15.6% |
| Low back pain | 6.8% | 23.2% | <0.5% | 5.0% | 21.9% | 11.1% | 30.0% |
| Malignant neoplastic disease | 7.7% | 4.4% | 4.5% | 9.4% | 15.8% | 10.8% | 18.4% |
| Osteoarthritis of knee | 4.6% | 7.7% | <0.5% | 2.0% | 14.8% | 5.1% | 11.4% |
| Osteoarthritis of hip | 3.0% | 0.5% | <0.5% | 1.1% | 5.1% | 2.6% | 2.8% |
| Peripheral vascular disease | 4.7% | 8.0% | 0.5% | 7.5% | 3.9% | 7.4% | 10.6% |
| Renal impairment | 21.1% | 1.8% | 7.6% | 37.5% | 13.1% | 37.2% | 32.9% |
| Venous thrombosis | 2.3% | 0.6% | 0.8% | 2.0% | 1.5% | 6.6% | 4.0% |
| Viral hepatitis | 2.3% | 3.9% | 0.6% | 2.5% | 1.7% | 2.6% | 5.9% |

Exact proportions have not been reported where counts were <10. Conditions were identified over the year prior and up to, and including, day of hospitalisation (note, HM and PHD had no prior time requirement with conditions primarily recorded on the day of hospital admission).
*CUIMC* Columbia University Irving Medical Center, *HIRA* Health Insurance Review & Assessment, *HM* HM Hospitales, *PHD* Premier Healthcare Database, *SIDIAP* The Information System for Research in Primary Care, *UC HDC* University of Colorado Health Data Compass, *VA OMOP* Department of Veterans Affairs, *COPD* chronic obstructive pulmonary disease, *GERD* gastroesophageal reflux disease.

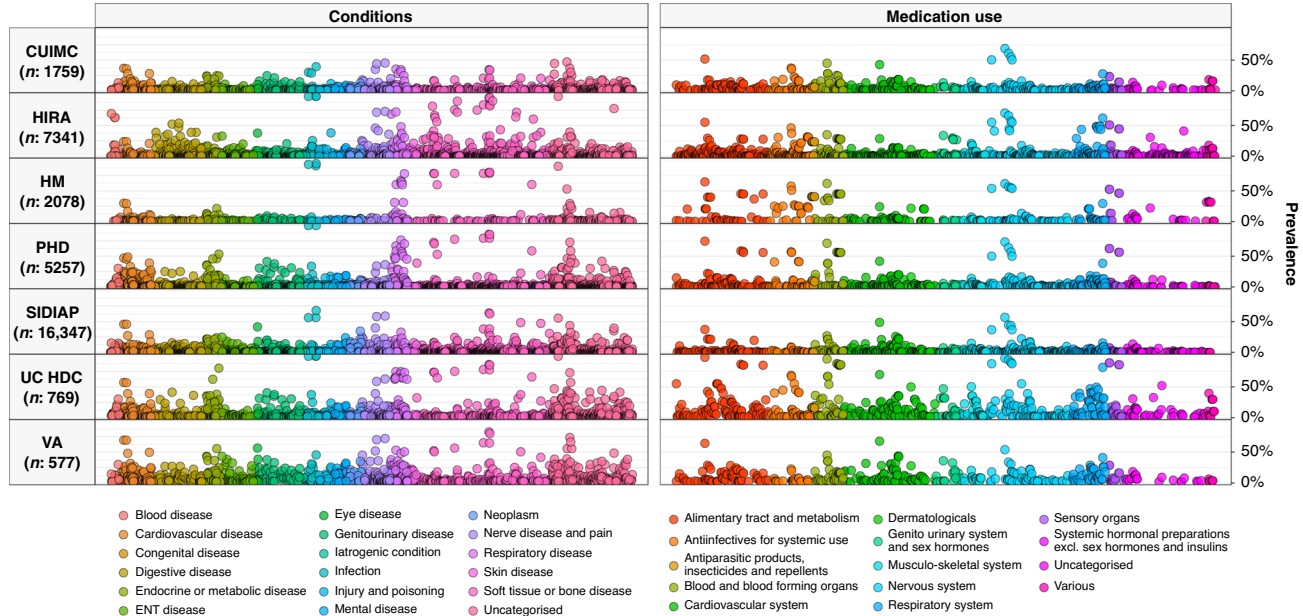

**Fig. 2 Prevalence of conditions and medication use among COVID-19 patients.** Individuals hospitalised with COVID-19 between December 2019 and April 2020. Conditions from up to a year prior, medication use from day of hospitalisation. Each dot represents one of these covariates with the colour indicating the type of condition/medication. CUIMC: Columbia University Irving Medical Center; HIRA: Health Insurance Review & Assessment; HM: HM Hospitales; PHD: Premier Healthcare Database; SIDIAP: The Information System for Research in Primary Care; UC HDC: University of Colorado Health Data Compass; VA OMOP: Department of Veterans Affairs.

The comparison with influenza made in our study adds important context when considering the characteristics of those hospitalised with COVID-19. Individuals hospitalised with COVID-19 appear to be more likely younger and male, and in the United States and Spain, to have fewer comorbidities than those hospitalised with influenza in previous years. Indeed, those

**Table 2 Prior medications of individuals hospitalised with COVID-19.**

| | CUIMC (*n*: 1759) | HIRA (*n*: 7341) | HM (*n*: 2078) | PHD (*n*: 5257) | SIDIAP (*n*: 16,347) | UC HDC (*n*: 769) | VA OMOP (*n*: 577) |
|---|---|---|---|---|---|---|---|
| *Antineoplastic and immunomodulating agents* | | | | | | | |
| Year up to, and including day of, hospitalisation | 9.6% | 4.5% | – | – | 3.4% | 12.9% | 13.2% |
| 30 days up to, and including day of, hospitalisation | 6.1% | 2.6% | – | – | 2.7% | 9.6% | 7.1% |
| Day of hospitalisation | 4.9% | 2.0% | 3.8% | 2.1% | 2.5% | 8.2% | 4.9% |
| *Agents acting on the renin–angiotensin system* | | | | | | | |
| Year up to, and including day of, hospitalisation | 27.3% | 15.9% | – | – | 30.2% | 39.8% | 49.6% |
| 30 days up to, and including day of, hospitalisation | 18.0% | 14.2% | – | – | 26.6% | 33.0% | 38.5% |
| Day of hospitalisation | 15.9% | 13.4% | 9.9% | 7.4% | 25.7% | 31.5% | 33.1% |
| *Antiepileptics* | | | | | | | |
| Year up to, and including day of, hospitalisation | 19.0% | 9.1% | – | – | 10.6% | 27.2% | 33.4% |
| 30 days up to, and including day of, hospitalisation | 9.9% | 5.2% | – | – | 8.1% | 21.6% | 22.7% |
| Day of hospitalisation | 9.1% | 4.2% | 3.3% | 11.3% | 7.7% | 20.2% | 16.6% |
| *Anti-inflammatory and antirheumatic products* | | | | | | | |
| Year up to, and including day of, hospitalisation | 23.6% | 63.9% | – | – | 30.9% | 52.9% | 46.1% |
| 30 days up to, and including day of, hospitalisation | 10.5% | 30.0% | – | – | 9.9% | 38.6% | 16.8% |
| Day of hospitalisation | 7.8% | 17.7% | 3.8% | 7.9% | 6.3% | 35.1% | 9.9% |
| *Antithrombotic agents* | | | | | | | |
| Year up to, and including day of, hospitalisation | 44.9% | 34.6% | – | – | 25.7% | 72.0% | 46.1% |
| 30 days up to, and including day of, hospitalisation | 29.9% | 17.1% | – | – | 21.9% | 67.5% | 16.8% |
| Day of hospitalisation | 28.3% | 11.4% | 38.2% | 38.4% | 21.3% | 66.8% | 9.9% |
| *Beta-blocking agents* | | | | | | | |
| Year up to, and including day of, hospitalisation | 26.9% | 10.1% | – | – | 13.8% | 32.8% | 44.4% |
| 30 days up to, and including day of, hospitalisation | 16.8% | 6.5% | – | – | 12.4% | 24.7% | 36.0% |
| Day of hospitalisation | 15.2% | 5.6% | 5.7% | 17.8% | 11.9% | 23.8% | 31.0% |
| *Calcium channel blockers* | | | | | | | |
| Year up to, and including day of, hospitalisation | 23.5% | 16.7% | – | – | 12.4% | 24.3% | 35.0% |
| 30 days up to, and including day of, hospitalisation | 15.0% | 14.6% | – | – | 10.5% | 20.5% | 27.0% |
| Day of hospitalisation | 13.4% | 14.0% | 6.8% | 9.7% | 10.1% | 19.9% | 24.3% |
| *Diuretics* | | | | | | | |
| Year up to, and including day of, hospitalisation | 27.6% | 10.2% | – | – | 24.6% | 33.3% | 42.6% |
| 30 days up to, and including day of, hospitalisation | 19.2% | 7.5% | – | – | 21.5% | 27.4% | 33.1% |
| Day of hospitalisation | 17.7% | 6.7% | 7.6% | 14.1% | 20.7% | 26.4% | 28.4% |
| *Drugs for acid related disorders* | | | | | | | |
| Year up to, and including day of, hospitalisation | 35.4% | 68.8% | – | – | 30.4% | 53.6% | 52.0% |
| 30 days up to, and including day of, hospitalisation | 20.6% | 36.5% | – | – | 22.7% | 44.5% | 36.2% |
| Day of hospitalisation | 17.9% | 29.3% | 39.7% | 21.8% | 21.5% | 42.4% | 28.9% |
| *Immunosuppressants* | | | | | | | |
| Year up to, and including day of, hospitalisation | 5.6% | 2.0% | – | – | 2.3% | 7.5% | 5.7% |
| 30 days up to, and including day of, hospitalisation | 3.9% | 0.7% | – | – | 1.9% | 6.1% | 3.5% |
| Day of hospitalisation | 3.7% | 0.4% | 2.5% | 1.5% | 1.7% | 5.9% | 2.6% |
| *Insulins and analogues* | | | | | | | |
| Year up to, and including day of, hospitalisation | 17.8% | 3.7% | – | – | 5.5% | 31.9% | 22.9% |
| 30 days up to, and including day of, hospitalisation | 10.9% | 2.7% | – | – | 4.9% | 26.9% | 18.4% |

**Table 2 (continued)**

| | CUIMC (*n*: 1759) | HIRA (*n*: 7341) | HM (*n*: 2078) | PHD (*n*: 5257) | SIDIAP (*n*: 16,347) | UC HDC (*n*: 769) | VA OMOP (*n*: 577) |
|---|---|---|---|---|---|---|---|
| Day of hospitalisation | 9.9% | 2.4% | <0.5% | 13.1% | 4.7% | 25.4% | 14.6% |
| *Lipid-modifying agents* | | | | | | | |
| Year up to, and including day of, hospitalisation | 33.8% | 18.4% | – | – | 26.8% | 41.7% | 61.9% |
| 30 days up to, and including day of, hospitalisation | 20.8% | 15.5% | – | – | 23.6% | 36.4% | 50.8% |
| Day of hospitalisation | 18.7% | 14.1% | 5.4% | 19.9% | 23.0% | 35.6% | 43.8% |

Study participants from HM and PHD were not required to have a year of prior history and so only medications at the time of hospitalisation are summarised.
*CUIMC* Columbia University Irving Medical Center, *HIRA* Health Insurance Review & Assessment, *HM* HM Hospitales, *PHD* Premier Healthcare Database, *SIDIAP* The Information System for Research in Primary Care, *UC HDC* University of Colorado Health Data Compass, *VA OMOP* Department of Veterans Affairs.

hospitalised with COVID-19 were consistently seen to be less likely to have COPD, cardiovascular disease and dementia than those hospitalised with influenza in recent years.

This study has also added important information on medication use by individuals hospitalised with COVID-19, based on electronic health records and claims data. There is tremendous interest in the risks and benefits of medications such as ACE inhibitors and ARBs for COVID-19, and whether other medications, such as ibuprofen, should be avoided. However, to date, there has been little evidence as to what proportion of those hospitalised with COVID-19 have previously been taking such medications. Our findings shed light on this area, and highlight the importance of further research on the benefits and harms associated with continued use of such treatments, especially those that are commonly taken amongst individuals with COVID-19. It has been seen here, for example, that between 1 in 10 and 2 in 5 of those hospitalised with COVID-19 were taking medicines acting on the renin–angiotensin system in the month before their hospital admission. The consequences of temporarily discontinuing such treatments on cardiovascular risks and mortality remain unknown[11]. Interestingly, corticosteroid use, recently shown to be effective to treat COVID-19[12], was consistently seen to be less prevalent in patients hospitalised with COVID-19 compared to those hospitalised with influenza across databases, as were alphablockers which some have hypothesised to have a beneficial effect in COVID-19[13].

It should be noted that the characteristics of individuals with COVID-19 have been described in this study at the particular point in time of admission to hospital. While this is of particular interest given its intrinsic link with healthcare utilisation, this only provides a snapshot of the whole picture. Those testing positive for COVID-19 in the community without progressing to hospitalisation can be expected to be younger and with fewer comorbidities than those hospitalised[9,14], while those progressing to intensive care can be expected to be older and in worse general health[3,15]. In addition, those being referred to or admitted to intensive care also seem more likely to be male[3,15]. Admission to hospital (and intensive care) is also influenced by a range of supply-side factors, such as availability of beds and criteria for admission, and so the characteristics of those hospitalised do not necessarily only reflect the characteristics accompanying severe illness. These factors likely explain some of the heterogeneity seen in those hospitalised with COVID-19 in this study. Geographic variation in populations and transmission dynamics can also be expected to explain the variation across sites. This particularly relevant for patients from South Korea, where given the management of COVID-19 in the country, the patient population is more closely reflective of those infected during early outbreaks.

**Study limitations**. The study was based on routinely collected data and so, as always, data quality issues must be considered. For instance, individuals were considered as having COVID-19 at the time of hospitalisation only if they had a test result or diagnosis indicating the disease, which will have led to the omission of individuals who can be suspected to have had the disease but lacking confirmation of it. Medical conditions may have been underestimated as they were based on the presence of condition codes, with the absence of such a record taken to indicate the absence of disease. Meanwhile, medication records indicate that an individual was prescribed or dispensed a particular drug, but this does not necessarily mean that an individual took the drug as originally prescribed or dispensed. Our study could be subject to exposure misclassification with false positives if a patient had a medication dispensing event but did not ingest the drug, but may also be subject to false negatives for non-adherent patients who continued their medication beyond the days of supply due to stockpiling. Medication use estimates based on the data collected at the time of hospitalization is particularly sensitive to misclassification, and may conflate baseline concomitant drug history with immediate treatment upon admission.

Time periods were inclusive of the day of hospitalisation, so that information on conditions captured on the day of hospitalisation and medication use up to and including the day of hospitalisation were captured. Other time frames could though have been considered for information prior to hospitalisation. Moreover, outcomes after the index date have not been summarised as these were outside the scope of this particular study. Associations between particular patient characteristics and the risks of hospitalisation and outcomes following hospitalisation were also beyond the scope of this study, and remain an area for further research.

Comparisons of individuals hospitalised with COVID-19 with individuals previously hospitalised with influenza have limitations. In particular, observed differences may be explained by changes in clinical practice or data capture procedures over time, rather than by differences in the individuals themselves. This is likely particularly relevant for comparisons of medication use which, in particular, can be expected to vary over time. Changes in practice in response to the COVID-19 pandemic, such as referrals which may not be observed in our data and thresholds for hospitalisation, also may in part explain some of the observed differences in patient profiles.

**Interpretation**. Rates of comorbidities and medication use are high among individuals hospitalised with COVID-19. Those individuals hospitalised with COVID-19 do not, however, appear to be in worse general health than those typically hospitalised with influenza. Indeed, in many cases, individuals hospitalised

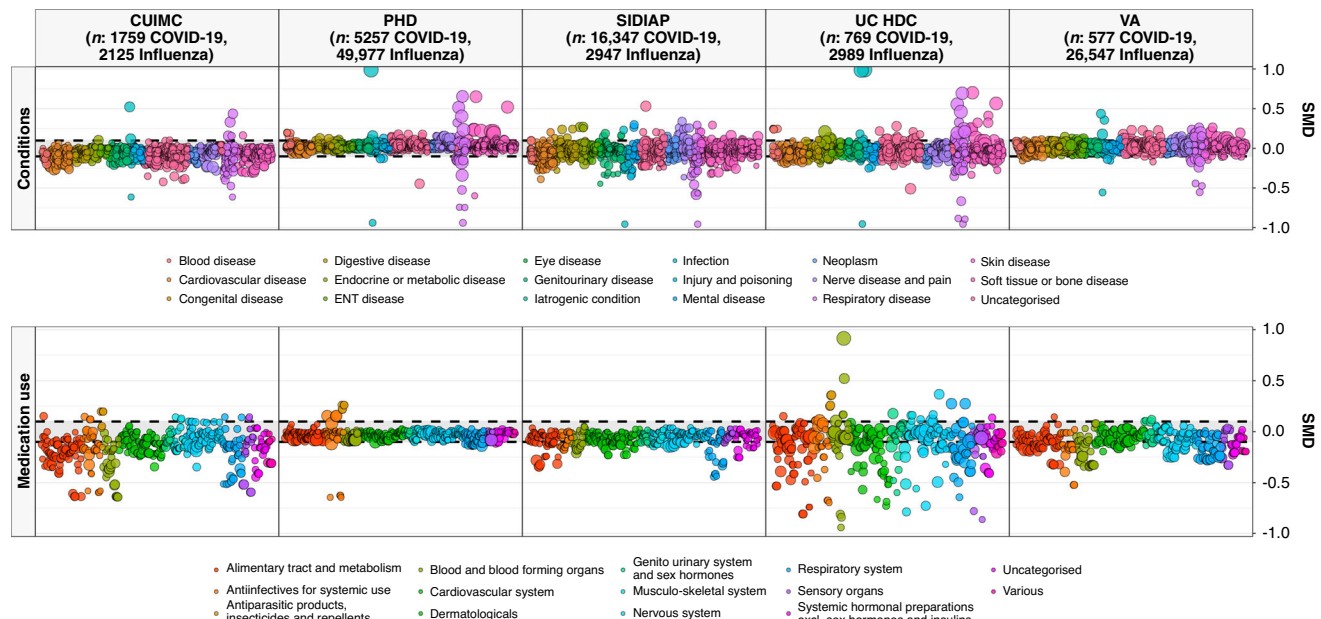

**Fig. 3 Standardised mean difference in conditions (top) and medication use (bottom) among COVID-19 patients compared to 2014–2019 influenza patients.** Individuals hospitalised with COVID-19 between December 2019 and April 2020 compared with those hospitalised with influenza between September 2014 and April 2019. Conditions from up to a year prior, medication use from day of hospitalisation. Each dot represents one of these covariates with the colour indicating the type of condition/medication and the size of the dot reflecting the prevalence of the variable in the COVID-19 study populations. CUIMC: Columbia University Irving Medical Center; PHD: Premier Healthcare Database; SIDIAP: The Information System for Research in Primary Care; VA OMOP: Department of Veterans Affairs.

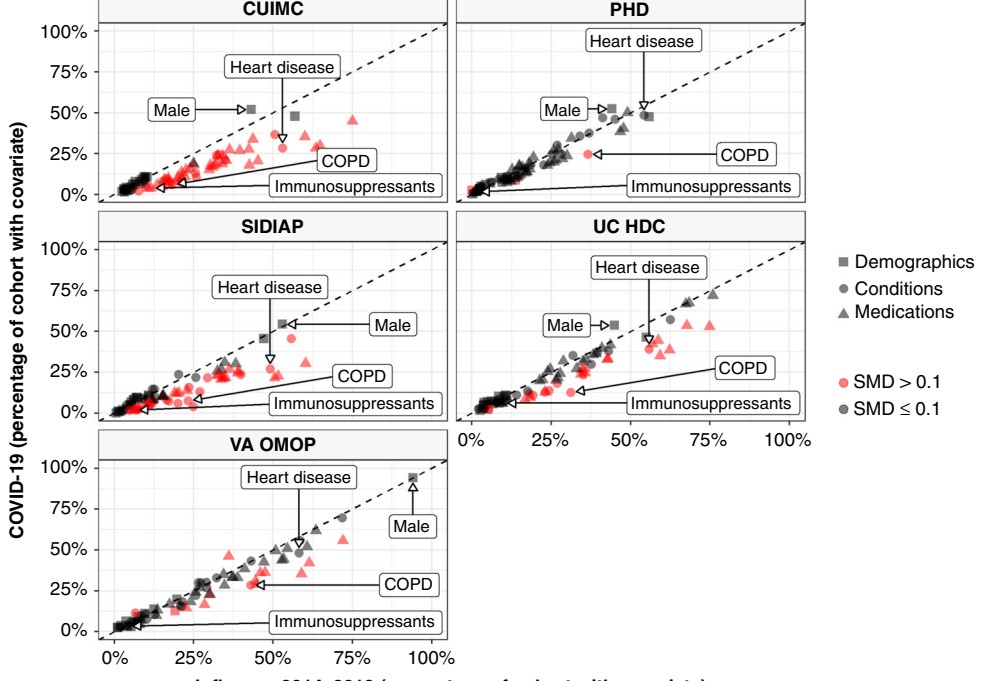

**Fig. 4 Characteristics of COVID-19 patients compared to 2014–2019 Influenza patients.** The plot compares demographics (age and sex), conditions (recorded over the year prior and up to the day of hospitalisation), and medications (1) from a year prior up to the day of hospitalisation, (2) from 30 days prior up to the day of hospitalisation and (3) on day of hospitalisation. Each dot represents one of these covariates with the colour indicating the absolute value of the standardised mean difference (SMD), with a SMD above 0.1 taken to indicate a difference in the prevalence of a particular covariate. The proportion male, with heart disease, with chronic obstructive pulmonary disease (COPD), and taking immunosuppressants (over the 30 days prior up to hospitalisation) are shown for illustration. CUIMC: Columbia University Irving Medical Center; PHD: Premier Healthcare Database; SIDIAP: The Information System for Research in Primary Care; UC HDC: University of Colorado Health Data Compass; VA OMOP: Department of Veterans Affairs.

with COVID-19 were seen to be younger and healthier than patients hospitalised with seasonal influenza. Patients hospitalised for COVID-19 are also more likely to be male in comparison to those hospitalised with influenza. Protecting those groups known to be vulnerable to influenza is likely to be a useful starting point to minimize the number of hospital admissions needed for COVID-19, but such strategies may need to be broadened so as to reflect the particular characteristics of individuals seen here to have been hospitalised with COVID-19.

## Methods

**Study design.** This is a cohort study based on routinely collected primary care and hospital electronic health records (EHRs), hospital billing data, and insurance claims data from the US, South Korea and Spain. The data sources used were mapped to the Observational Medical Outcomes Partnership (OMOP) Common Data Model (CDM)[16]. The open-science Observational Health Data Sciences and Informatics (OHDSI) network maintains the OMOP CDM, and its members have developed a wide range of tools to facilitate analyses of such mapped data[17]. Two particular benefits of this approach were that contributing centres did not need to share patient-level data and common analytical code could be applied across databases.

**Data sources.** Data from the US, South Korea, and Spain underpinned the study. EHR data from the US came from the Columbia University Irving Medical Center (CUIMC), covering NewYork-Presbyterian Hospital/Columbia University Irving Medical Center, University of Colorado Health Data Compass (UC HDC), which includes the UCHealth System with data from 12 hospitals, and United States Department of Veterans Affairs (VA OMOP), which includes 170 medical centres. In addition, data from a US hospital billing system database came from the Premier Hospital database (PHD). EHR data from Spain came from The Information System for Research in Primary Care (SIDIAP), a primary care records database that covers ~80% of the population of Catalonia, Spain[18], and the inpatient care database of HM Hospitales (HM), a hospital group which includes 15 general hospitals from all over Spain, with detailed hospital admission information for COVID-19 patients from March 1 to April 20, 2020. Data from South Korea came from Health Insurance Review & Assessment (HIRA), a repository of national claims data which is collected in the process of reimbursing healthcare providers[19]. In addition, the analysis was also performed on US EHR data from Tufts-Clinical Academic Research Enterprise Trust (CLARET), which covers Tufts Medical Center (TMC), and the STAnford medicine Research data Repository (STARR-OMOP), which includes data from Stanford Health Care[20], and on data from the Daegu Catholic University Medical Center, a teaching hospital in Daegu, South Korea, covered by Federated E-health Big Data for Evidence Renovation Network (FEEDER-NET). These latter sites are not reported here due to smaller study populations, but the results are reported in the accompanying interactive website (http://evidence.ohdsi.org/Covid19CharacterizationHospitalization/).

Each database was mapped to the OMOP CDM, which was developed as a means of standardising the structure, content and semantics of observational databases[21]. The OHDSI network has maintained and further developed this common data model since 2014, developing tools to both facilitate the mapping of source data to the OMOP CDM and analysing it once mapped. While the journey of implementing the extract, transform, and load process varies across sites (given factors such as infrastructure, size of the database, the complexity of the mapping, and the technical expertise available), the destination is the same for each site; a database which conforms to the requirements of the OMOP CDM. The mapping of source codes to standard concepts is done using OMOP standardised vocabularies, which are regularly updated[22]. In OHDSI studies, such as this one, data partners run the same analysis package using a distributed network approach, where analyses are run locally and only aggregated summary statistics are then shared. OHDSI held a COVID-19 studyathon in March 2020, during which this study was initiated. To note, the OMOP CDM has also been used to facilitate network COVID-19 studies by the 4CE consortium, in which trajectories of laboratory test measurements among COVID-19 patients were described[23], and is being used by the N3C consortium to help harmonise EHR data on COVID-19 patients[24].

**Study participants.** Patients hospitalised between December 2019 and April 2020 with COVID-19 were identified on the basis of having a hospitalisation along with a confirmatory diagnosis or test result of COVID-19 within a time window from 21 days prior to admission up to the end of their hospitalisation. This time window was chosen so as to include those who had the diagnosis made prior to their hospitalisation and allow for a delay in test results or diagnoses to be made or recorded. Patients were also required to be aged 18 years or older at the time of hospitalisation. The same algorithm was used to identify COVID-19 cases across sites, except for CUIMC where the algorithm was adjusted to account for local coding practice. The codes used to identify COVID-19 are described in more detail in Supplementary Methods.

Analogous criteria were used for identifying individuals hospitalised with influenza between September 2014 and April 2019, with individuals identified on the basis of having a hospitalisation along with a confirmatory diagnosis or test result of influenza within a time window from 21 days prior to admission up to the end of their hospitalisation. For SIDIAP, influenza cases were only available up to the end of 2017. An additional cohort of those hospitalised with influenza between September 2009 and April 2010 was also identified. The motivation for this latter group was that the 2009–2010 flu epidemic included many cases of A(H1N1) pdm09 infection, which had different clinical characteristics and associated severity compared to the seasonal flu. Each individual's first hospitalisation with COVID-19 or a particular flu season was considered.

Except for the HM and Premier databases, individuals were required to have a minimum of 365 days of prior observation time available for the primary analysis, to allow for a comprehensive capture of baseline diagnoses and medications up to their hospitalisation. Individuals' observation period in the OMOP CDM reflects a period during which any clinical event that happens to the patient is expected to be recorded. The way this is operationalised does though depend on the type of source data being used. For claims data, for example, observation periods are generally inferred from the enrolment periods to a particular plan, while for EHR observation periods generally begin at the time of interaction with the health system. Consequently, as the requirement for prior observation time could exclude persons with little prior healthcare utilisation or without sustained health insurance, we also characterised cohorts without this restriction in a sensitivity analysis. Given the nature of data collection, individuals in HM and Premier had no prior observation time available and so no requirement for prior observation time was imposed.

**Patient features and data analysis.** Age at hospitalisation and sex distributions were summarised. Medication use was calculated over three time periods: (1) from a year prior up to, and including, the day of hospitalisation, (2) from 30 days prior up to, and including, the day of hospitalisation and (3) the day of hospitalisation. Only the latter time period was reported for HM and Premier. Drug eras were calculated to give the span of time when an individual is assumed to be exposed to a particular active ingredient. These begin on the start date of the first drug exposure and end on the observed end date if available, or were inferred (for example, based on the number of days of supply). A persistence window of up to 30 days was permitted between two medication records for them to be considered as part of the same drug era. Individual medications were categorised using Anatomical Therapeutic Chemical (ATC) groupings. All drugs are reported in full in a dedicated interactive website (see 'Results' section), but specific classes are reported here based on recent interest due to their potential effects (positive or negative) on COVID-19 susceptibility or severity: agents acting on the renin–angiotensin system (including angiotensin-converting enzyme (ACE) inhibitors and angiotensin II receptor blockers (ARBs)), antiepileptics, beta-blocking agents, calcium channel blockers, diuretics, drugs for acid related disorders, immunosuppressants, insulins, analogues and lipid-modifying agents (such as statins). Prevalence of medication use for each time window was determined by the proportion of persons who had at least one day during the time window over-lapping with a drug era for each medication or drug class of interest. Conditions were identified on the basis of SNOMED codes, with all descendent codes included. Similarly, all recorded diagnoses are available for consultation in the accompanying website, but a list of key conditions is reported here based on recent reports of associations with COVID-19 infection or outcomes.

Age distributions in each cohort were plotted. The proportion of a cohort having a particular characteristic was described, with the prevalence of all medications and conditions captured in the databases depicted using Manhattan-style plots. Standardised mean differences (SMD) were calculated when comparing characteristics of study cohorts, with a SMD of above 0.1 generally considered to indicate a meaningful difference in the prevalence of a covariate[25]. The SMDs for all variables were plotted in Manhattan-style plots. In addition, the prevalence of particular conditions or medications of interest among those hospitalised with COVID-19 (Y axis) were compared to those hospitalised with influenza (X axis) in scatter plots, with dots on the top-left indicating a higher prevalence among those hospitalised with COVID-19 and dots on the bottom-right indicating a higher prevalence among those hospitalised with influenza.

**Ethical approval.** All the data partners received Institutional Review Board (IRB) approval or exemption. STARR-OMOP had approval from IRB Panel #8 (RB-53248) registered to Leland Stanford Junior University under the Stanford Human Research Protection Program (HRPP). The use of VA data was reviewed by the Department of Veterans Affairs Central Institutional Review Board (IRB) and was determined to meet the criteria for exemption under Exemption Category 4(3) and approved the request for Waiver of HIPAA Authorization. The research was approved by the Columbia University Institutional Review Board as an OHDSI network study. The IRB number for use of HIRA data was AJIB-MED-EXP-20-065. HM Hospitales and SIDIAP analyses were approved by the Clinical Research Ethics Committee of the IDIAPJGol (project code: 20/070-PCV). The UC-HDC data use was reviewed by Colorado Multi-Institutional Review Board (COMIRB) and was determined to meet the criteria for exemption under Exemption Category

4(3) and approved the request for Waiver of HIPAA Authorization (protocol # 20-0730).

**Reporting summary**. Further information on research design is available in the Nature Research Reporting Summary linked to this article.

## Data availability

The aggregated results set, that does not include patient-level health information, is available via http://evidence.ohdsi.org/Covid19CharacterizationHospitalization/. Data partners (Columbia University Irving Medical Center [CUIMC], Health Insurance Review & Assessment [HIRA], HM Hospitales [HM], Premier Hospital database [PHD], The Information System for Research in Primary Care [SIDIAP], United States Department of Veterans Affairs [VA OMOP] and University of Colorado Health Data Compass [UC HDC]) contributing to this study remain custodians of individual patient-level health information.

## Code availability

The analytic code used in the study has been made available at https://github.com/ohdsi-studies/Covid19HospitalizationCharacterization[26]. This study package is run from R and is a OHDSI CohortDiagnostics-type package[27]. This makes use of other OHDSI packages, such as FeatureExtraction[28], to extract patient characteristics for user-specified cohorts in the common data model.

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

## Acknowledgements

This project has received support from the European Health Data and Evidence Network (EHDEN) project. EHDEN has received funding from the Innovative Medicines Initiative 2 Joint Undertaking (JU) under grant agreement No. 806968. The JU receives support from the European Union's Horizon 2020 research and innovation programme and EFPIA. This research received partial support from the National Institute for Health Research (NIHR) Oxford Biomedical Research Centre (BRC), US National Institutes of Health, US Department of Veterans Affairs, Janssen Research & Development and IQVIA. This work was also supported by the Bio Industrial Strategic Technology Development Program (20001234) funded by the Ministry of Trade, Industry & Energy (MOTIE, Korea) and a grant from the Korea Health Technology R&D Project through the Korea Health Industry Development Institute (KHIDI), funded by the Ministry of Health & Welfare, Republic of Korea [grant number: HI16C0992]. Personal funding included Versus Arthritis [21605], Medical Research Council Doctoral Training Partnership (MRC-DTP) [MR/K501256/1] (J.C.E.L.); Medical Research Council (MRC) and Fundación Alfonso Martín Escudero (FAME) (A.P.U.); Innovation Fund Denmark (5153-00002B) and the Novo Nordisk Foundation (NNF14CC0001) (B.S.K.H.); VINCI [VA HSR RES 13-457] (S.L.D., M.E.M. and K.E.L.); NIHR Senior Research Fellowship (SRF-2018-11-ST2-004, D.P.A.); Bill & Melinda Gates Foundation (INV-016201); the Intramural Research Program of the National Institutes of Health/National Library of Medicine/Lister Hill National Center for Biomedical Communications (V.H.); and the Direcció General de Recerca i Innovació en Salut from the Department of Health of the Generalitat de Catalunya. No funders had a direct role in this study. The views and opinions expressed are those of the authors and do not necessarily reflect those of the Clinician Scientist Award programme, NIHR, Department of Veterans Affairs or the United States Government, NHS or the Department of Health, England. The authors appreciate the Korean Health Insurance Review and Assessment Service for providing data and HM Hospitales for making their data publicly available as part of the COVID Data Save Lives project. UC HDC is supported by the Health Data Compass Data Warehouse project (healthdatacompass.org). V.H. work was supported by the Intramural Research Program, National Institute of Health.

## Author contributions

Study conception: E.B., S.C.Y., L.S., D.V., A.G., F.N., D.P.V. and P.R.; data analysis: E.B., A.S., S.C.Y., T.F., S.D. and J.P.; initial drafting of the paper: E.B., D.P.V. and P.R. All authors (E.B., S.C.Y., A.G.S., K.K., H.A., M.T.F.A., A.A., H.A., O.A., T.M.A., M.A., C.A, J. M.B., J.C., A.C.C., A.D., F.J.D., T.D.S., S.D., T.F., S.F.B., W.G., A.G., J.H., G.H., V.H., H.J., Y.J., C.Y.J., B.S.K.H., D.K., S.K., Y.K., S.K., J.C.E.L., H.L., K.E.L., R.M., M.E.M., P.P.M., D. R.M., K.N., F.N., A.O., R.W.P., J.P., J.D.P., A.P.U., G.R., C.R., Y.R., P.R., L.M.S., M.S., N. H.S., A.S., S.S., M.S., M.A.S., J.N.S., D.V., S.V., H.W., A.E.W., B.B.Y., L.Z., O.Z., D.P.A. and P.R.) were involved in the design of the work, interpretation of the data and revising the draft paper.

## Competing interests

All authors have completed the ICMJE uniform disclosure form, with the following declarations made: D.P.A. reports grants and other from AMGEN, grants, non-financial support and other from UCB Biopharma, grants from Les Laboratoires Servier, outside the submitted work; and Janssen, on behalf of IMI-funded EHDEN and EMIF consortiums, and Synapse Management Partners have supported training programmes organised by DPA's department and open for external participants. D.V. reports personal fees from Bayer, outside the submitted work, and he is a full-time employee at a pharmaceutical company. DM reports funding support from the Wellcome Trust, NIHR, Scottish CSO and Tenovus Scotland for research unrelated to this work. S.C.Y. reports grants from Korean Ministry of Health & Welfare, grants from Korean Ministry of Trade, Industry & Energy, during the conduct of the study. A.G. reports personal fees from Regeneron Pharmaceuticals, outside the submitted work, and she is a full-time employee at Regeneron Pharmaceuticals. This work was not conducted at Regeneron Pharmaceuticals. Y.J. reports employee of AbbVie and owns company stock. A.A. reports that he is currently employed at Alberta Health Services (AHS) as a Data Science Lead redeployed as an epidemiologist to aid in the COVID-19 response. This work was not conducted at AHS, within AHS working hours, or with AHS staff. He contributed and conducted this work as an Independent Epidemiologist, as a member of the Observational Health Data Sciences and Informatics (OHDSI) Network. P.R. reports grants from Innovative Medicines Initiative, grants from Janssen Research and Development, during the conduct of the study. M.S. reports grants from US National Science Foundation, grants from US National Institutes of Health, grants from IQVIA, personal fees from Janssen Research and Development, during the conduct of the study. G.H. reports grants from US NIH National Library of Medicine, during the conduct of the study; grants from Janssen Research, outside the submitted work. A.P.U. reports grants from Fundacion Alfonso Martin Escudero, grants from Medical Research Council, outside the submitted work. H.A. reports personal fees from Eli Lilly and Company, outside the submitted work. A.S. reports personal fees from Janssen Research & Development, during the conduct of the study; personal fees from Janssen Research & Development, outside the submitted work. A.S. is a full-time employee of Janssen and shareholder of Johnson & Johnson. G.R. is a full-time employee of Janssen and shareholder of Johnson & Johnson. F.D. reports personal fees from Janssen Research & Development, during the conduct of the study; personal fees from Janssen Research & Development, outside the submitted work. R.W.P. reports grants from Korean Ministry of Health & Welfare, grants from Korean Ministry of Trade, Industry & Energy, during the conduct of the study. J.P. reports grants from Korean Ministry of Health & Welfare, grants from Korean Ministry of Trade, Industry & Energy, during the conduct of the study. J.C. reports grants from Korean Ministry of Health & Welfare, grants from Korean Ministry of Trade, Industry & Energy, during the conduct of the study. S.D. reports grants from Anolinx, LLC, grants from Astellas Pharma, Inc, grants from AstraZeneca Pharmaceuticals LP, grants from Boehringer Ingelheim International GmbH, grants from Celgene Corporation, grants from Eli Lilly and Company, grants from Genentech Inc., grants from Genomic Health, Inc., grants from Gilead Sciences Inc., grants from GlaxoSmithKline PLC, grants from Innocrin Pharmaceuticals Inc., grants from Janssen Pharmaceuticals, Inc., grants from Kantar Health, grants from Myriad Genetic Laboratories, Inc., grants from Novartis International AG, grants from Parexel International Corporation through the University of Utah or Western Institute for Biomedical Research outside the submitted work. H.J.

reports grants from Korean Ministry of Health & Welfare, grants from Korean Ministry of Trade, Industry & Energy, during the conduct of the study. B.S.K.H. reports grants from Innovation Fund Denmark (5153-00002B) and the Novo Nordisk Foundation (NNF14CC0001), outside the submitted work. K.K. reports she is an employee of IQVIA. CR reports he is an employee of IQVIA. J.S. reports other from Janssen R&D, during the conduct of the study; other from Janssen R&D, outside the submitted work; and J.S. was a full-time employee of Johnson & Johnson, or a subsidiary, at the time the study was conducted. J.S. owns stock, stock options, and pension rights from the company. R.M. reports and is an employee of Janssen Research and Development. W.G. is an AbbVie employee. P.R. reports and is an employee of Janssen Research and Development and shareholder of Johnson & Johnson. M.S. is a full-time employee of Janssen R&D, and a shareholder of Johnson & Johnson. J.H. reports other from Janssen Research & Development, during the conduct of the study; other from Janssen Research & Development, outside the submitted work; and full-time employee of Janssen and shareholder of Johnson & Johnson. All other authors declare no competing interests.

## Additional information

Edward Burn [1,2,51], Seng Chan You [3,51], Anthony G. Sena [4,5], Kristin Kostka [6], Hamed Abedtash [7], Maria Tereza F. Abrahão [8], Amanda Alberga [9], Heba Alghoul [10], Osaid Alser [11], Thamir M. Alshammari [12], Maria Aragon [1], Carlos Areia [13], Juan M. Banda [14], Jaehyeong Cho [3], Aedin C. Culhane [15], Alexander Davydov [16,17], Frank J. DeFalco [4], Talita Duarte-Salles [1], Scott DuVall [18,19], Thomas Falconer [20], Sergio Fernandez-Bertolin [1], Weihua Gao [21], Asieh Golozar [22,23], Jill Hardin [4], George Hripcsak [20,24], Vojtech Huser [25], Hokyun Jeon [26], Yonghua Jing [21], Chi Young Jung [27], Benjamin Skov Kaas-Hansen [28,29], Denys Kaduk [16,30], Seamus Kent [31], Yeesuk Kim [32], Spyros Kolovos [33], Jennifer C. E. Lane [33], Hyejin Lee [34], Kristine E. Lynch [18,19], Rupa Makadia [4], Michael E. Matheny [35,36], Paras P. Mehta [37], Daniel R. Morales [38], Karthik Natarajan [20,24], Fredrik Nyberg [39], Anna Ostropolets [20], Rae Woong Park [3,26], Jimyung Park [26], Jose D. Posada [40], Albert Prats-Uribe [2], Gowtham Rao [4], Christian Reich [6], Yeunsook Rho [33], Peter Rijnbeek [5], Lisa M. Schilling [41], Martijn Schuemie [4,42], Nigam H. Shah [40], Azza Shoaibi [4], Seokyoung Song [43], Matthew Spotnitz [20], Marc A. Suchard [42], Joel N. Swerdel [4], David Vizcaya [44], Salvatore Volpe [20],

Haini Wen[45], Andrew E. Williams [46], Belay B. Yimer [47], Lin Zhang [48,49], Oleg Zhuk[16], Daniel Prieto-Alhambra [2 ✉] & Patrick Ryan[4,50]

[1]Fundació Institut Universitari per a la recerca a l'Atenció Primària de Salut Jordi Gol i Gurina (IDIAPJGol), Barcelona, Spain. [2]Centre for Statistics in Medicine (CSM), Nuffield Department of Orthopaedics, Rheumatology and Musculoskeletal Sciences (NDROMS), University of Oxford, Oxford, UK. [3]Department of Biomedical Informatics, Ajou University School of Medicine, Suwon, Korea. [4]Janssen Research and Development, Titusville, NJ, USA. [5]Department of Medical Informatics, Erasmus University Medical Center, Rotterdam, The Netherlands. [6]Real World Solutions, IQVIA, Cambridge, MA, USA. [7]Eli Lilly and Company, Indianapolis, IN, USA. [8]Faculty of Medicine, University of Sao Paulo, Sao Paulo, Brazil. [9]Observational Health Data Sciences and Informatics Network, Alberta, Canada. [10]Faculty of Medicine, Islamic University of Gaza, Gaza, Palestine. [11]Massachusetts General Hospital, Harvard Medical School, Boston, MA, USA. [12]Medication Safety Research Chair, King Saud University, Riyadh, Saudi Arabia. [13]Nuffield Department of Clinical Neurosciences, University of Oxford, Oxford, UK. [14]Department of Computer Science, Georgia State University, Atlanta, GA, USA. [15]Data Science, Dana-Farber Cancer Institute. Biostatistics, Harvard TH Chan School of Public Health, Boston, MA, USA. [16]Odysseus Data Services, Inc., Cambridge, MA, USA. [17]Department for Microbiology, Virology and Immunology, Belarusian State Medical University, Minsk, Belarus. [18]Department of Veterans Affairs, Salt Lake City, UT, USA. [19]University of Utah School of Medicine, Salt Lake City, UT, USA. [20]Department of Biomedical Informatics, Columbia University, New York, NY, USA. [21]Health Economics and Outcomes Research, AbbVie, North Chicago, IL, USA. [22]Pharmacoepidemiology, Regeneron, NY, USA. [23]Department of Epidemiology, Johns Hopkins School of Public, Baltimore, MD, USA. [24]New York-Presbyterian Hospital, New York, NY, USA. [25]National Library of Medicine, National Institutes of Health, Bethesda, MD, USA. [26]Department of Biomedical Sciences, Ajou University Graduate School of Medicine, Suwon, Korea. [27]Division of Respiratory and Critical Care Medicine, Department of Internal Medicine, Daegu Catholic University Medical Center, Daegu, Korea. [28]Clinical Pharmacology Unit, Zealand University Hospital, Køge, Denmark. [29]NNF Centre for Protein Research, University of Copenhagen, København, Denmark. [30]Department of Pediatrics № 2, V. N. Karazin Kharkiv National University, Kharkiv, Ukraine. [31]Science Policy and Research, National Institute for Health and Care Excellence, London, UK. [32]Department of Orthopaedic Surgery, College of Medicine, Hanyang University, Seoul, Korea. [33]Nuffield Department of Orthopaedics, Rheumatology and Musculoskeletal Sciences (NDROMS), University of Oxford, Oxford, UK. [34]Bigdata Department, Health Insurance Review & Assessment Service, Wonju, Korea. [35]GRECC, Tennessee Valley Healthcare System VA, Nashville, TN, USA. [36]Department of Biomedical Informatics, Vanderbilt University Medical Center, Nashville, TN, USA. [37]College of Medicine-Tucson, University of Arizona, Tucson, AZ, USA. [38]Division of Population Health and Genomics, University of Dundee, Dundee, UK. [39]School of Public Health and Community Medicine, Institute of Medicine, Sahlgrenska Academy, University of Gothenburg, Gothenburg, Sweden. [40]Department of Medicine, School of Medicine, Stanford University, Stanford, CA, USA. [41]Data Science to Patient Value Program, Department of Medicine, University of Colorado Anschutz Medical Campus, Aurora, CO, USA. [42]Department of Biostatistics, UCLA Fielding School of Public Health, University of California, Los Angeles, CA, USA. [43]Department of Anesthesiology and Pain Medicine, Catholic University of Daegu, School of Medicine, Gyeongsan, Korea. [44]Bayer Pharmaceuticals, Barcelona, Spain. [45]Shuguang Hospital affiliated to Shanghai University of Traditional Chinese Medicine, Shanghai, China. [46]Tufts Institute for Clinical Research and Health Policy Studies, Boston, MA, USA. [47]Centre for Epidemiology Versus Arthritis, Manchester Academic Health Science Centre, The University of Manchester, Manchester, UK. [48]School of Public Health, Peking Union Medical College, Chinese Academy of Medical Sciences, Beijing, China. [49]Melbourne School of Population and Global Health, The University of Melbourne, Melbourne, Australia. [50]Columbia University, New York, NY, USA. [57]These authors contributed equally: Edward Burn, Seng Chan You. ✉email: daniel.prietoalhambra@ndorms.ox.ac.uk

