## [Peer Review File · Nature Communications]

REVIEWER COMMENTS

Reviewer #1 (Remarks to the Author):

This study leveraged the OHDSI network to characterize the demographics and prior conditions and medications in 34,128 patients admitted for COVID-19 at seven hospitals in three countries. The study was also performed at several other hospitals, but the results from these sites were not included in this paper due to small sample sizes. The main finding is that COVID-19 patients, compared to patients hospitalized for influenza in prior years, were more male, younger, and healthier when they were admitted. (Of note, the study does not look at clinical course, treatment, or outcomes--just patient characteristics at admission.) Though, I think of greater significance is that this paper demonstrates the potential of large amounts of observational data, such as in the OHDSI network, to study COVID-19. It is one of the biggest COVID-19 cohorts I've seen; their sites span multiple continents; they have a mix of EHR and claims data; and, having comparison groups, like influenza patients from prior years, differentiates them from groups looking only at COVID-19. The federated model used in this study, where sites run analyses locally, rather than combining data on patients with COVID-19 into a central repository, avoids regulatory barriers and enables rapid analyses on large scales.

As acknowledged by the authors, there are always data quality issues with routinely-collected data. This gets amplified when working with data from multiple hospital systems. So, most of my comments/questions are to clarify details in their methods.

- 1) The aggregate data used for this study are all available on the accompanying website (<http://evidence.ohdsi.org:3838/Covid19CharacterizationHospitalization/>). The main data table contains 26,774 variables across the different sites. However, only 100 variables are shown per page. I would have to click through 268 pages to get all the data. Can the authors add a link to download the entire dataset (e.g., as a CSV file)?
- 2) Patients with COVID-19 and influenza were identified with a "confirmatory diagnosis or test result". What specific codes and tests were used? In particular, how were COVID-19 patients back in December through February identified; and, how did they handle the new diagnosis and laboratory tests that were regularly introduced during March and April? Were lab tests for COVID-19 recorded as LOINC codes at sites, or was local mapping needed? Were COVID-19 tests flagged as positive in the source EHRs, or did they have to parse result notes to determine which patients were positive?
- 3) The algorithm for selecting study participants was "adjusted to account for local coding practice" at CUIMC. What does that mean? How was selecting patients at CUIMC different?
- 4) What does "a minimum of 365 days of prior observation time" mean for EHR data? Is that selecting patients who, for example, had an encounter/visit recorded at least a year prior to COVID-19 admission?
- 5) I'm confused about whether the prior 365 and prior 30 day windows include the day of admission. Page 4 indicates that it includes the hospitalization. However, other places in the text have "prior to hospitalisation", "prior and up to the day of hospitalisation", and "prior to the day of hospitalisation". It would make more sense to me if the 365 and 30 day windows ended the day before hospitalization, so you can distinguish conditions and medications that happened before hospitalization and the ones that happened for the first time as part of the COVID-19 hospitalization. For example, were the reported conditions patients' co-morbidities, or are they sequelae of COVID-19? Were the patients already taking the medications, or did their physicians give them the drugs when they arrived at the hospital to treat COVID-19?

6) I know that some hospitals are receiving many COVID-19 patients through referrals. Is it possible that COVID-19 patients appear healthier than influenza patients because they are more likely to receive their usual care at another institution?

7) Can the authors explain the data mapping process and workflow? For example, how were local codes mapped to SNOMED and ATC? I see a combination of SQL and R code in the GitHub. What part of the analysis was SQL on the OMOP databases versus analyses in R? How did sites send in their aggregate results for final analysis? Were small counts masked, and did that impact the findings?

8) There are 7 data sites, but the author list contains 50 affiliations. I'm very interested in the broader context of this study. How did this group come together? How is it organized? What are their plans? What did all those authors and institutions do to contribute to this work? Just a few sentences would be useful. Typically, this information might not be too important. However, as I noted above, I think the future potential of this network is more important than the results in this one paper. I expect many more COVID-19 studies to come out of OHDSI, and this model could be used to study future pandemics. It would be good to understand the "secret sauce" that made this study possible.

9) The Figure 3 caption states that color indicates the standard mean difference. However, I think it actually corresponds to the condition and medication category.

10) The abstract states that this study summarized the "entire medical histories" of patients. However, the paper only presents demographics, conditions and medications. It does not include laboratory tests, procedures, prior visits, etc.

Reviewed by Griffin M Weber, MD, PhD.

Reviewer #2 (Remarks to the Author):

The authors conducted a plain descriptive analysis of demographics, previously recorded conditions, and medication use of hospitalized patients with COVID-19, compared with hospitalized patients with influenza. While the joint efforts of different research groups to put together are appreciated, I would expect an in-depth comparison between the hospitalized patients with COVID-19 and those without COVID-19 (i.e. with influenza or with no respiratory diseases), which was currently missing in the manuscript. Please see below for my comments and suggestions:

Major comments:

1. When hospitalized patients with COVID-19 were compared with those with influenza, no statistical models or tests were applied to conduct a multivariate comparison. For example, the authors found hospitalized patients with COVID-19 were slightly younger with comparable or lower prevalence of comorbidities. But it was not clear whether these two factors were confounded in the dataset and whether these age and prevalence of comorbidities were statistically different among COVID-19 and flu hospitalizations.
2. It was not clear what are the objectives of this study. If the primary aim were to characterize in detail the demographics and medical histories of COVID-19 hospitalizations, it would be helpful to see the three-way comparisons among COVID-19 hospitalizations, influenza hospitalizations and non-respiratory-infection hospitalizations. However, only direct comparison between COVID-19 patients and influenza patients were available.
3. If the primary aim were to look for associations between demographics, previously recorded conditions and medications and COVID-19 hospitalizations, the authors did not attempt to conduct any association analysis or estimate the potential burden of COVID-19 hospitalizations from this

dataset.

Minor comments:

1. The abstract was not structured in a format for publication in Nature Communications.
2. The proportion of female patients was higher in South Korea probably because there were a few very big clusters of COVID-19 cases involving many female infections, e.g. the Shincheonji outbreak

Reviewer #1 (Remarks to the Author):

This study leveraged the OHDSI network to characterize the demographics and prior conditions and medications in 34,128 patients admitted for COVID-19 at seven hospitals in three countries. The study was also performed at several other hospitals, but the results from these sites were not included in this paper due to small sample sizes. The main finding is that COVID-19 patients, compared to patients hospitalized for influenza in prior years, were more male, younger, and healthier when they were admitted. (Of note, the study does not look at clinical course, treatment, or outcomes--just patient characteristics at admission.) Though, I think of greater significance is that this paper demonstrates the potential of large amounts of observational data, such as in the OHDSI network, to study COVID-19. It is one of the biggest COVID-19 cohorts I've seen; their sites span multiple continents; they have a mix of EHR and claims data; and, having comparison groups, like influenza patients from prior years, differentiates them from groups looking only at COVID-19. The federated model used in this study, where sites run analyses locally, rather than combining data on patients with COVID-19 into a central repository, avoids regulatory barriers and enables rapid analyses on large scales.

As acknowledged by the authors, there are always data quality issues with routinely-collected data. This gets amplified when working with data from multiple hospital systems. So, most of my comments/questions are to clarify details in their methods.

1) The aggregate data used for this study are all available on the accompanying website (<http://evidence.ohdsi.org:3838/Covid19CharacterizationHospitalization/>). The main data table contains 26,774 variables across the different sites. However, only 100 variables are shown per page. I would have to click through 268 pages to get all the data. Can the authors add a link to download the entire dataset (e.g., as a CSV file)?

➔ **We have added a download button below the tables that allows the user to download a corresponding csv file.**

2) Patients with COVID-19 and influenza were identified with a "confirmatory diagnosis or test result". What specific codes and tests were used? In particular, how were COVID-19 patients back in December through February identified; and, how did they handle the new diagnosis and laboratory tests that were regularly introduced during March and April? Were lab tests for COVID-19 recorded as LOINC codes at sites, or was local mapping needed? Were COVID-19 tests flagged as positive in the source EHRs, or did they have to parse result notes to determine which patients were positive?

➔ **As part of the process of mapping to the OMOP common data model, sites mapped their source data to standard concepts used in the common data model. When identifying cases of COVID-19 on the basis of diagnostic codes we used the mapped standard concepts. However, for testing results we identified cases on the basis of both standard concepts and specific source concepts because of difficulty in finding suitable standard codes in the common model to represent all the source testing codes being used to represent COVID-19 tests and their results.**

➔ **We have added a further appendix with details of all the codes used, along with links to an OHDSI ATLAS website that provides the definitions used (with both summary descriptions, along with the specific SQL code used).**

3) The algorithm for selecting study participants was "adjusted to account for local coding practice" at CUIMC. What does that mean? How was selecting patients at CUIMC different?

➔ **Each site reviewed the suitability of the definitions used to identify COVID-19 cases. For CUIMC it was seen that some generic coronavirus codes, such as "Novel coronavirus infection", would lead to the inclusion of non-COVID-19 patients. These codes were therefore not included for CUIMC, and we have noted these omitted codes in the appendix.**

4) What does "a minimum of 365 days of prior observation time" mean for EHR data? Is that selecting patients who, for example, had an encounter/visit recorded at least a year prior to COVID-19 admission?

➔ **Individuals with 365 days of prior observation time have an observation window that started 365 days or more prior to their index date, i.e. they have been present in the database for at least a year prior to their hospitalisation with COVID-19.**

➔ **The convention for the common data model is that, as a general assumption, during an observation period any clinical event that happens to the patient is expected to be recorded.**

➔ **For claims data, observation periods are inferred from the enrolment periods to a particular health benefit plan. Meanwhile, as noted, for EHR data the observation period cannot be determined explicitly, and so typically an interaction with the health provider will inform an individual's start of observation.**

➔ **We have clarified this in the text (lines 189-195). To note, we also performed a sensitivity analysis, where no requirement for prior history was imposed and the results from this were seen to be similar to the primary analysis.**

5) I'm confused about whether the prior 365 and prior 30 day windows include the day of admission. Page 4 indicates that it includes the hospitalization. However, other places in the text have "prior to hospitalisation", "prior and up to the day of hospitalisation", and "prior to the day of hospitalisation". It would make more sense to me if the 365 and 30 day windows ended the day before hospitalization, so you can distinguish conditions and medications that happened before hospitalization and the ones that happened for the first time as part of the COVID-19 hospitalization. For example, were the reported conditions patients' co-morbidities, or are they sequelae of COVID-19? Were the patients already taking the medications, or did their physicians give them the drugs when they arrived at the hospital to treat COVID-19?

➔ **These all include day of hospitalisation (individual's baseline), and we have made this clearer throughout the paper.**

➔ **The decision to include information captured on individual's day of hospitalisation was so as to include information on medical history that would not have otherwise been identified in prior data (i.e. with the observation, and coding, of individuals comorbidities when admitted).**

➔ **This does mean that the full dataset captures both information relating to chronic conditions and relating to COVID-19. In the paper we have focused predominantly on the former. Of note, even limiting to data from the day prior and earlier would not necessarily**

have separated the two completely, given potential outpatient visits relating to COVID-19 prior to hospitalisation.

- ➔ For medications, we included day zero in each (i.e rather than -365 to -30, it was -365 to 0) so that each were relative to baseline. This allowed, for example, of a description of the use of agents acting on the renin-angiotensin system in CUIMC showing that 27% of individuals had used them in the prior year up to and including day of hospitalisation, 18% in the prior 30 days and 16% on the day of hospitalisation. We believe this framing is useful for readers interested in the proportion of those hospitalised who are users of a certain medication.
- ➔ Using information collected on the day of hospitalisation also allowed for the inclusion of HM and Premier, both of which did not have prior history typically available. As can be seen from these, important information on chronic conditions was captured in these data sources.
- ➔ We do, however, fully agree that there are various different windows during which information could have been extracted, and have noted this as a potential limitation (391-397).

6) I know that some hospitals are receiving many COVID-19 patients through referrals. Is it possible that COVID-19 patients appear healthier than influenza patients because they are more likely to receive their usual care at another institution?

- ➔ In the study we have only included an individual's first observed hospitalisation with COVID-19, and generally required that individuals had a year of prior observation time.
- ➔ Where an individual was observed to have multiple hospitalisations only the first would be included. However, it is possible that some individuals may have been previously seen in another hospital not captured in our database. This though is likely to be of more relevance for EHR data than for claims data.
- ➔ We have added a comment on this in the limitations section of the discussion (line 401-404)

7) Can the authors explain the data mapping process and workflow? For example, how were local codes mapped to SNOMED and ATC? I see a combination of SQL and R code in the GitHub. What part of the analysis was SQL on the OMOP databases versus analyses in R? How did sites send in their aggregate results for final analysis? Were small counts masked, and did that impact the findings?

- ➔ While the common data model used across sites is uniform, the implementation of the extract, transform, and load (ETL) process to convert data to the CDM varies across sites because of factors such as infrastructure, size of the database, the complexity of the ETL, and the technical expertise available. Mapping source concepts to standard codes makes use of OHDSI standard vocabularies. The approach used for ETL and mapping source codes have been described in detail in "The Book of OHDSI" (<https://ohdsi.github.io/TheBookOfOhdsi/>). We have added a reference to this resource, and added further information in the text (lines 156 to 167).
- ➔ The analysis itself makes use of OHDSI methods library, in particular the OHDSI R CohortDiagnostics. This extracts the summary characteristics for user-defined cohorts, directly querying the database using other OHDSI packages. We have added further details to the text (lines 232-235), with the added references containing detailed documentation on the underlying packages used.

8) There are 7 data sites, but the author list contains 50 affiliations. I'm very interested in the broader context of this study. How did this group come together? How is it organized? What are their plans? What did all those authors and institutions do to contribute to this work? Just a few sentences would be useful. Typically, this information might not be too important. However, as I noted above, I think the future potential of this network is more important than the results in this one paper. I expect many more COVID-19 studies to come out of OHDSI, and this model could be used to study future pandemics. It would be good to understand the "secret sauce" that made this study possible.

- ➔ **In March, 2020, the OHDSI community organised a 'studyathon', with more than 290 people from 29 different countries participating. This study was one of the studies started during this time. This study was conceptualised during this period, and subsequently we have worked together to implement the study (finalising the study package, running across sites, and working to summarise the results in the form of this manuscript). The diversity in the authorship list reflects the various perspectives and skills required in order to implement this particular study.**
- ➔ **The study has also benefitted tremendously from the prior work done by members of the OHDSI network, particularly in developing the common data model over the past ten years and a range of tools to aid in data analyses. This existing infrastructure has meant that this large network study of COVID-19 could be performed in a timely and robust manner.**
- ➔ **We have added this further context to the study to the manuscript (lines 156-167)**

9) The Figure 3 caption states that color indicates the standard mean difference. However, I think it actually corresponds to the condition and medication category.

- ➔ **Thank you for spotting this. This has now been corrected.**

10) The abstract states that this study summarized the "entire medical histories" of patients. However, the paper only presents demographics, conditions and medications. It does not include laboratory tests, procedures, prior visits, etc.

- ➔ **We have removed the word 'entire', and updated the abstract in line with the requirements for Nature Communications.**

Reviewed by Griffin M Weber, MD, PhD.

Reviewer #2 (Remarks to the Author):

The authors conducted a plain descriptive analysis of demographics, previously recorded conditions, and medication use of hospitalized patients with COVID-19, compared with hospitalized patients with influenza. While the joint efforts of different research groups to put together are appreciated, I would expect an in-depth comparison between the hospitalized patients with COVID-19 and those without COVID-19 (i.e. with influenza or with no respiratory diseases), which was currently missing in the manuscript. Please see below for my comments and suggestions:

- ➔ **We have clarified our research objectives in the paper. The first was to summarise the characteristics of those hospitalised with COVID-19. The second objective was to compare the characteristics of those hospitalised with influenza in prior years.**
- ➔ **Our analytic approach was designed to address these specific research questions. For the first, we have provided a vast range of aggregated summary statistics on more than 34,000 patients hospitalised with COVID-19. We believe this information is informative in itself, providing both details not well captured in other datasets (such as historic medication use) and a breadth not seen in other studies (with information on patients from the US, South Korea, and Spain provided). For the second research question, we compared characteristics across study cohorts using standardised mean differences (SMDs). The univariate comparisons were made using SMDs as p-values would, given the size of the data, often be <0.05 even with only small differences in the percentages. With our research questions descriptive in nature, further regression modelling would not be in accordance with our particular questions.**
- ➔ **We believe this comparison of COVID-19 with influenza is particularly relevant, given the implications for public health strategies. Comparisons to other populations are, unfortunately, beyond the scope of this study.**

Major comments:

1. When hospitalized patients with COVID-19 were compared with those with influenza, no statistical models or tests were applied to conduct a multivariate comparison. For example, the authors found hospitalized patients with COVID-19 were slightly younger with comparable or lower prevalence of comorbidities. But it was not clear whether these two factors were confounded in the dataset and whether these age and prevalence of comorbidities were statistically different among COVID-19 and flu hospitalizations.

- ➔ **We have compared the characteristics of those hospitalised with COVID-19 to those hospitalised with influenza using SMDs. We have added our rationale for using SMDs to the text (lines 224-225), with a SMDs of more than 0.1 taken to indicate a difference in the prevalence of a covariate between groups. The use of SMDs is particularly advantageous in the context of this study, where sample sizes are particularly large (p-values alone would often be <0.05 even with only small differences in the percentages).**
- ➔ **We have also clarified the research questions addressed in this study, which are descriptive in nature (lines 117-121). As such confounding, a concept intrinsically linked to causal inference not descriptive epidemiology, has not been considered.**

2. It was not clear what are the objectives of this study. If the primary aim were to characterize in detail the demographics and medical histories of COVID-19 hospitalizations, it would be helpful to

see the three-way comparisons among COVID-19 hospitalizations, influenza hospitalizations and non-respiratory-infection hospitalizations. However, only direct comparison between COVID-19 patients and influenza patients were available.

- ➔ **We have clarified our research questions (lines 117-121), and we believe the comparison with influenza is particularly informative.**
- ➔ **It was beyond the scope of this particular study to compare the characteristics of those hospitalised with COVID-19 to other patient populations.**

3. If the primary aim were to look for associations between demographics, previously recorded conditions and medications and COVID-19 hospitalizations, the authors did not attempt to conduct any association analysis or estimate the potential burden of COVID-19 hospitalizations from this dataset.

- ➔ **We have clarified the objectives of our study (lines 115 to 121). The first objective was to summarise the characteristics of individuals hospitalised with COVID-19, and the second was to compare the characteristics of these individuals with those previously hospitalised with influenza.**
- ➔ **For the first objective we extracted many aggregated summary statistics. For the second we have used standardised mean differences (SMDs) to compare characteristics between those hospitalised with influenza. We have added further rationale for the use of SMDs to compare characteristics between the two cohorts in each of the database (lines 223-225)**
- ➔ **Assessing associations between exposures and risk of hospitalisation and outcomes following hospitalisation was outside of the scope of this study. We have added this as a potential limitation (lines 391-393), and added this as an area for further research.**
- ➔ **We believe that the results of this descriptive study are of interest in themselves. We have summarised the characteristics from over 34,000 patients hospitalised with COVID-19 from the US, South Korea, and Spain. We have also provided a comparison with patients previously hospitalised with influenza, adding important context. These findings can help provide both an improved understanding of the profiles of individuals being hospitalised with COVID-19 and underpin future research, such as by providing a starting point for selecting the characteristics of interest for more detailed association studies.**

Minor comments:

1. The abstract was not structured in a format for publication in Nature Communications.

- ➔ **We have updated the abstract in line with the requirements for Nature Communications**

2. The proportion of female patients was higher in South Korea probably because there were a few very big clusters of COVID-19 cases involving many female infections, e.g. the Shincheonji outbreak.

- ➔ **We agree that the South Korea data is, given the management of COVID-19 in the country, is particularly reflective of the nature of early outbreaks. We have added a comment on this to the text (lines 371-375).**

REVIEWERS' COMMENTS

Reviewer #1 (Remarks to the Author):

The reviewers addressed all the questions I had about their methods. In particular:

- 1) The full results table is now available as a single download on the website.
- 2) The authors clarified how they defined the different time periods, and they added some discussion about limitations and alternatives.
- 3) An Appendix was added listing the codes used to select their COVID-19 cohort. (It contains links to a "full description" of the logic. Though, these are protected by a login.) A very helpful paragraph was added to the Data Source section, which describes the process in OHDSI of mapping local codes to the OMOP CDM, running analyses locally, and sharing aggregated summary statistics. Reference to the OHDSI studyathon is useful in understanding the origin of this project.

Minor Comment:

- 1) Two related papers were published this week. The 4CE Consortium used a similar approach (multiple hospitals ran local analyses and shared aggregate statistics) to characterize laboratory test trajectories of patients with COVID-19 [Nature Digital Medicine, <https://doi.org/10.1038/s41746-020-00308-0>]. Some of the 4CE sites used OMOP as their data source. N3C is also using OMOP data from multiple hospitals and tools from OHDSI to study COVID-19, but by copying hospitals' patient-level data into a centralized secure data enclave [Journal of the American Medical Informatics Association, <https://doi.org/10.1093/jamia/ocaa196>]. This is completely optional, but the authors might want to mention some of these ongoing complementary efforts to give some broader context of this study.

Reviewer #2 (Remarks to the Author):

The authors have addressed all my comments.

REVIEWERS' COMMENTS

Reviewer #1 (Remarks to the Author):

The reviewers addressed all the questions I had about their methods. In particular:

- 1) The full results table is now available as a single download on the website.
- 2) The authors clarified how they defined the different time periods, and they added some discussion about limitations and alternatives.
- 3) An Appendix was added listing the codes used to select their COVID-19 cohort. (It contains links to a "full description" of the logic. Though, these are protected by a login.) A very helpful paragraph was added to the Data Source section, which describes the process in OHDSI of mapping local codes to the OMOP CDM, running analyses locally, and sharing aggregated summary statistics. Reference to the OHDSI studyathon is useful in understanding the origin of this project.

Minor Comment:

- 1) Two related papers were published this week. The 4CE Consortium used a similar approach (multiple hospitals ran local analyses and shared aggregate statistics) to characterize laboratory test trajectories of patients with COVID-19 [Nature Digital Medicine, <https://doi.org/10.1038/s41746-020-00308-0>]. Some of the 4CE sites used OMOP as their data source. N3C is also using OMOP data from multiple hospitals and tools from OHDSI to study COVID-19, but by copying hospitals' patient-level data into a centralized secure data enclave [Journal of the American Medical Informatics Association, <https://doi.org/10.1093/jamia/ocaa196>]. This is completely optional, but the authors might want to mention some of these ongoing complementary efforts to give some broader context of this study.

- **We have added these references, which we agree help to contextualise our study**

Reviewer #2 (Remarks to the Author):

The authors have addressed all my comments.